# Isoleucine Enhanced the Function of the Small Intestinal Mucosal Barrier in Weaned Piglets to Alleviate Rotavirus Infection

**DOI:** 10.3390/ani14213146

**Published:** 2024-11-02

**Authors:** Rongkun Zhao, Changsheng Jiang, Yuchen Yuan, Shen Zhang, Ahmed H. Ghonaim, Chuanyan Che, Xiaojin Li, Mengmeng Jin, Erhui Jin, Xiangfang Zeng, Shenghe Li, Man Ren

**Affiliations:** 1Anhui Provincial Key Laboratory of Animal Nutritional Regulation and Health, College of Animal Science, Anhui Science and Technology University, Fengyang 233100, China; rongkunzhao@126.com (R.Z.); jiangcs@ahstu.edu.cn (C.J.); 18855065915@163.com (Y.Y.); 15755797521@163.com (S.Z.); checy@ahstu.edu.cn (C.C.); lixj@ahstu.edu.cn (X.L.); jinmm@ahstu.edu.cn (M.J.); jineh@ahstu.edu.cn (E.J.); 2National Key Laboratory of Agricultural Microbiology, College of Animal Sciences and Veterinary Medicine, Huazhong Agricultural University, Wuhan 430070, China; a.ghonaim@webmail.hzau.edu.cn; 3Desert Research Center, Cairo 11435, Egypt; 4State Key Laboratory of Animal Nutrition, College of Animal Science and Technology, China Agricultural University, Beijing 100193, China; zengxf@cau.edu.cn

**Keywords:** isoleucine, rotavirus, small intestine, mucosal barrier, weaned piglet

## Abstract

Rotavirus is a major cause of diarrhea in young children and animals, especially piglets, leading to substantial economic losses in the global pig industry. Isoleucine, a branched-chain amino acid, plays an important role in regulating nutrient metabolism and has been shown to improve diarrhea. This study aimed to evaluate the effects of isoleucine supplementation on the mucosal immune barrier of the small intestine in rotavirus-infected weaned piglets. The results showed that the addition of isoleucine to the diet mitigated the effects of rotavirus infection on the intestinal morphology and the mucosal barrier, as well as the physiological function of weaned piglets, and improved the antioxidant and immune functions of these piglets to a certain extent. These findings offer valuable insights contributing to a deeper understanding of the role of isoleucine.

## 1. Introduction

Rotavirus (RV) is a major pathogen that causes acute enteritis in young children and animals, including nursing and weaned piglets [1,2,3]. It is globally prevalent, infecting many species, including humans [4], pigs, cattle [5], dogs [6], and sheep [7]. Some RV genotypes have also shown zoonotic potential [8,9]. Piglets are highly susceptible to rotavirus infection, which is associated with high mortality rates [10]. RV adversely impacts the health and growth of pigs in the global pig industry, especially in developing countries, resulting in significant economic losses [1,11]. Currently, rehydration (oral and intravenous) remains the main treatment strategy for rotavirus gastroenteritis (RVGE), and no specific drugs are available [12]. The virus is transmitted via the fecal–oral route and results in the destruction of mature small intestinal enterocytes, leading to villus atrophy [8,13], which subsequently causes malabsorption and diarrhea [14]. However, the effects of RV infection on the function and structure of the porcine intestinal mucosal barrier are not fully understood.

Isoleucine (Ile) is a branched-chain amino acid that is essential for humans and pigs, and it can regulate protein metabolism, fat deposition, and glucose utilization in the body [15,16]. A previous study has shown that Ile supplementation can alleviate acute diarrhea caused by malnutrition in children [17]. Additionally, Ile serves as an important energy substrate and is essential for immune protein synthesis, thereby improving immune function [18]. Previous studies have reported that Ile can improve growth performance, feed intake, feed utilization, relative gut length, and intestinal morphology in hybrid catfish [19]. Furthermore, it can regulate the intestinal immune function by increasing the excretion of β-defensins via the activation of the Sirt1/ERK/90RSK signaling pathway [20,21]. Thus, isoleucine has the potential to prevent pathogen invasion by boosting immunity.

In recent years, the potential of nutrients to prevent and treat viral infections has attracted the attention of many scientists, given their multiple beneficial effects on the host, such as maintaining normal intestinal function, protecting the intestinal tract from pathogenic bacteria, and modulating immunity [22,23,24]. However, studies on the effects of dietary Ile on the intestinal tract of enteropathogenic virus-infected piglets are limited. Given the multiple beneficial effects of Ile, the present study aimed to preliminarily investigate its effects on the intestinal tissue structure, digestive capacity, and immune function in RV-infected weaned piglets.

## 2. Materials and Methods

### 2.1. Animal Ethics Statement

All animal experiments were approved by the Animal Ethics Committee of Anhui Science and Technology University, under protocol number AK2023013. All experimental procedures were carried out in strict accordance with the “Guidelines for the Care and Use of Test Animals” of Anhui Province.

### 2.2. Animal Experiment

The rotavirus used in this study was a tissue-culture-adapted Ohio State University strain (ATCC#VR-893). RV preparation and virus titer determination [tissue culture infective dose 50 (TCID_50_)] were conducted as described previously [25].

Forty-eight healthy Duroc × Landrace × Yorkshire weaned barrows (6.88 ± 0.54 kg) at 21 days old were randomly divided into 6 groups (eight replicates in each group). Each piglet was individually housed in a metabolic cage with a self-feeder and a nipple watering device, in rooms with moderate temperature control.

Piglets had free access to feed and water during the trial period. The control (CON) and RV groups received the basal diet, while the 0.5% Ile and RV + 0.5% Ile groups were fed the basal diet containing 0.5% Ile. The 1% Ile and RV + 1% Ile groups were fed a basal diet containing 1% Ile. On the 15th day of feeding, all piglets were orogastrically inoculated with 5 mL of 100 mmol/L sodium bicarbonate (NaHCO_3_) solution. Twenty minutes later, the infected group was orogastrically inoculated with 5 mL of RV (1 × 10^6^ TCID_50_/mL), and the uninfected group was orogastrically inoculated with 5 mL of 0.9% sodium chloride (NaCl) solution. At 5 days post infection (dpi), all piglets were euthanized, and samples were collected. The animal experimental design and treatments are shown in Table 1.

The basal diets met the NRC (National Research Council) standard nutritional requirements for 7–11 kg pigs [26]. In this study, 0, 0.5%, and 1% L-isoleucine was added to the basal diet. Alanine was included for isonitrogen treatment. The L-Ile and alanine in the diets replaced the corn starch in the basal diet in equal amounts. The specific ingredient composition and nutritional levels are shown in Table 2.

### 2.3. Sample Collection

At 5 dpi, all piglets were euthanized via intracardiac injection of Na pentobarbital (50 mg/kg of body weight) and exsanguinated. The abdomen was immediately opened and flushed with 0.9% NaCl solution. Approximately 2 cm from the midpoint of the duodenum, jejunum and ileum were collected, and the fatty tissue on the surface of the intestine was peeled off to avoid extrusion during the sampling process. The samples were then put into 4% paraformaldehyde fixative for preservation. Simultaneously, the intestinal tubes were cut longitudinally, and the contents were flushed out with 0.9% NaCl solution, and then, the mucosa of the duodenum, jejunum, and ileum were scraped and put into liquid nitrogen for preservation.

### 2.4. Intestinal Morphology

Paraffin-embedded fixed intestinal segments were sectioned into 5 μm sections. Sections were deparaffinized in xylene and then hydrated in a concentration gradient of 100%, 95%, 85%, and 70% alcohol. The sections were then stained with hematoxylin and eosin using an H&E stain kit (Solarbio, Beijing, China) according to the manufacturer’s protocol to assess tissue morphology. Examination of sections was performed at least in 10 well-oriented intact villi and their associated crypts using light microscopy. Goblet cells were stained using a periodic acid–Schiff (PAS) stain kit (Solarbio, Beijing, China) according to the manufacturer’s instructions. Villus height and crypt depth were measured, and the number goblet cells was counted by Image J 1.53a software.

### 2.5. Western Blot

The intestinal mucosal protein was extracted by RIPA lysis buffer with 1% PMSF (Beyotime, Haimen, China) at 4 °C. The protein concentration was measured using a BCA Protein Assay Kit (Thermo Fisher, Waltham, MA, USA). A total of 20 μg of protein from each sample was loaded onto a 10% SDS-PAGE gel and then transferred to a polyvinylidene difluoride (PVDF) membrane. To prevent nonspecific binding, the PVDF membranes were blocked using 5% bovine serum albumin (BSA) at room temperature for 2 h. Next, the membranes were incubated overnight at 4 °C with the appropriate primary antibody. Following incubation, the membranes were washed three times with Tris-buffered saline with Tween 20 (TBST) and then incubated with either HRP-linked goat anti-rabbit or HRP-linked goat anti-mouse antibodies at room temperature for 2 h. Visualization was performed using an ECL detection kit (Solarbio, Beijing, China). The antibodies used for the Western blot are listed in Table 3.

### 2.6. Digestive Enzyme and Antioxidant Activity Assessment

The activities of the intestinal mucosal lipase, amylase, sucrase, total antioxidant capacity (T-AOC), superoxide dismutase (SOD), and glutathione (GSH) were assessed using kits according to the manufacturer’s protocol. All kits were purchased from Nanjing Jiancheng Bioengineering Institute (Table 4).

### 2.7. Cytokine Index Assessment

The expressions of interferon-γ (IFN-γ), secretory immunoglobulin A (SIgA), and defensins (DEFβ) 1 and 2 were measured in the intestinal mucosa using kits following the manufacturer’s protocol. All kits were purchased from Beijing Dakome Co., Beijing, China (Table 5).

### 2.8. Statistical Analysis

Data were analyzed by two-way analysis of variance (ANOVA), with model factors including the effects of L-isoleucine administration, RV infection, and their interaction. When significant effects were present, the Tukey test was used to compare means amongst treatments with significant effects. Data are presented as means ± standard deviations (SDs). *p* values of <0.05 were considered statistically significant. All statistical analysis was performed using GraphPad Prism 8 software.

## 3. Results

### 3.1. Effects of Ile Supplementation on Small Intestinal Morphology in RV-Infected Piglets

To evaluate the effect of Ile on small intestinal morphology in RV-infected piglets, the histomorphology of the small intestine was assessed. As shown in Figure 1 and Table 6, compared with the CON group, 1% Ile increased the villus height (*p* < 0.05) and villus height/crypt depth ratio (*p* < 0.05) in the duodenum of RV-uninfected piglets. In contrast, compared with the CON group, RV infection significantly decreased the villus height/crypt depth ratio in the duodenum (*p* < 0.05), jejunum (*p* < 0.05), and ileum (*p* < 0.05), increased the crypt depth of the jejunum (*p* < 0.05), and obviously decreased the villus height of the ileum (*p* < 0.05). However, compared with the RV group, 1% Ile significantly increased the villus height (*p* < 0.05) and villus height/crypt depth ratio (*p* < 0.05) in the ileum of RV-infected piglets. These data suggest that Ile could ameliorate the damage caused by RV infection to the histological structure of the small intestine.

### 3.2. Effects of Ile Supplementation on Mucosal Barrier Function of the Small Intestine in RV-Infected Piglets

The primary function of goblet cells is to synthesize and secrete mucus, which plays an important role in inhibiting viral infection. Figure 2 and Table 7 show the periodic acid–Schiff (PAS) staining performed to observe the distribution and number of goblet cells in small intestinal tissue. The results showed that, compared with the CON group, RV infection decreased the number of goblet cells in the duodenum and ileum (*p* < 0.05). However, 0.5% and 1% Ile supplementation did not significantly affect the number of goblet cells in the three segments of the small intestine in RV-uninfected piglets (*p* > 0.05). Additionally, compared with the RV group, 1% Ile supplementation increased the number of goblet cells in the duodenum and ileum of RV-infected piglets (*p* < 0.05). Based on these results, it can be concluded that 1% Ile supplementation significantly improves the histological structure and increases the number of goblet cells in the small intestine of RV-infected piglets, while 0.5% Ile has minimal effect on RV-induced histological changes and goblet cell number. Therefore, the following studies focused solely on 1% Ile supplementation.

To further investigate the effect of isoleucine on the small intestinal mucosal barrier of RV-infected weaned piglets, Figure 3 and Table 8 illustrate the expression of the tight junction markers Claudin 1 and Occludin, as well as the mucin Muc 1, which is a component of intestinal mucus, determined by Western blot analysis. The results showed that compared with the CON group, RV infection significantly decreased the expression of Claudin 1 in the duodenum (*p* < 0.05) and jejunum (*p* < 0.05), decreased the expression of Occludin in the jejunum (*p* < 0.05), and decreased the expression of Muc 1 in both the duodenum (*p* < 0.05) and jejunum (*p* < 0.05). In addition, compared with the CON group, 1% Ile significantly increased the expression of Claudin 1 and Occludin in the duodenum (*p* < 0.05) and ileum (*p* < 0.05), as well as the expression of Muc 1 in the ileum (*p* < 0.05). Furthermore, compared with the RV group, 1% Ile increased the expression of Claudin 1 and Muc 1 in the duodenum (*p* < 0.05) and jejunum (*p* < 0.05) and enhanced the expression of Occludin in the jejunum (*p* < 0.05) of RV-infected piglets. These data indicate that Ile promotes the differentiation and mucus secretion of small intestinal goblet cells, improves intestinal epithelial tight junctions, and alleviates the impairment of small intestinal barrier function by RV infection to a certain extent.

### 3.3. Effects of Ile Supplementation on Digestive Enzymes of the Small Intestinal Mucosa in RV-Infected Piglets

Table 9 demonstrates the effects of dietary supplementation with Ile on the activity of digestive enzymes in the small intestinal mucosa in RV-infected piglets. Compared with the CON group, RV infection significantly decreased the activity of lipase (*p* < 0.05) in the duodenum, reduced the activity of amylase in the jejunum, and decreased the activity of sucrase in the duodenum, jejunum, and ileum (*p* < 0.05). Compared with the CON group, 1% Ile increased the activity of amylase and sucrase (*p* < 0.05) in the ileum of the RV-uninfected piglets. Additionally, compared with the RV group, 1% Ile could increase the activity of lipase (*p* < 0.05) in the duodenum and the activity of sucrase (*p* < 0.05) in the ileum of the RV-infected piglets. These findings demonstrate that Ile was able to partially reverse the RV-induced reductions in the activity of digestive enzymes in the small intestinal mucosa.

### 3.4. Effects of Ile Supplementation on Antioxidant Capacity of the Small Intestinal Mucosa in RV-Infected Piglets

Rotavirus infection can induce oxidative stress [27]; meanwhile, isoleucine possesses antioxidative properties [28]. Therefore, we speculated that Ile may mitigate the oxidative stress induced by RV. Table 10 depicts the effects of Ile on the antioxidant expression of the small intestinal mucosa in RV-infected piglets. Compared with the CON group, the expression of GSH (*p* < 0.05) was significantly decreased in the ileum of the RV group. Compared with the CON group, 1% Ile supplementation increased the expression of T-AOC (*p* < 0.05) in the jejunum and GSH (*p* < 0.05) in the ileum of RV-uninfected piglets. During viral infection, 1% Ile supplementation significantly increased the expression of SOD (*p* < 0.05) in the duodenum and GSH (*p* < 0.05) in the jejunum of RV-infected piglets compared with the RV group. These data suggest that Ile could ameliorate the reduced antioxidant expression of the small intestine in piglets caused by RV infection.

### 3.5. Effects of Ile Supplementation on Cytokines of Small Intestinal Mucosa in RV-Infected Piglets

During viral infection, the body enters an antiviral state by producing immune cytokines like interferons and immunoglobulins [29]. Notably, interferon gamma (IFN-γ) and secretory immunoglobulin A (SIgA) play important roles in intestinal mucosal immunity [30,31]. Table 11 shows the effects of Ile on the expression of IFN-γ and SIgA in the small intestinal mucosa of RV-infected piglets. Compared with the CON group, RV infection significantly increased the expression of IFN-γ (*p* < 0.05) in the jejunum and ileum. Compared with the RV group, 1% Ile supplementation increased the expression of IFN-γ (*p* < 0.05) in the duodenum of RV-infected piglets. Moreover, we found that the expression of SIgA in the RV+1% Ile group was higher than that of the other three groups, particularly in the duodenum (*p* < 0.05) and jejunum (*p* < 0.05) compared with the RV group. These findings reveal that Ile could modulate the small intestinal mucosal immune function following RV infection in weaned piglets. 

### 3.6. Effects of Ile Supplementation on Defensins of the Small Intestinal Mucosa in RV-Infected Piglets

Previous studies have reported that beta defensins play an important role in resisting multiple viral infections [32,33,34]. In this study, ELISA kits were used to evaluate the expression of porcine beta defensins 1 and 2 in the mucosa of the duodenum, jejunum, and ileum. As shown in Table 12, compared with the CON group, RV infection significantly decreased the expression of DEFβ1 (*p* < 0.05) in the jejunum and decreased the expression of DEFβ2 in the duodenum (*p* < 0.05). Additionally, compared with the CON group, 1% Ile supplementation increased the expression of DEFβ1 (*p* < 0.05) in the ileum of RV-uninfected piglets. Furthermore, compared with the RV group, 1% Ile significantly increased the expression of DEFβ1 (*p* < 0.05) in the jejunum and increased the expression of DEFβ2 (*p* < 0.05) in duodenum of RV-infected piglets. These data suggest that Ile could modulate the defensin expression in the small intestinal mucosa of weaned piglets following RV infection.

## 4. Discussion

Isoleucine, an essential functional branched-chain amino acid, plays a vital role in the regulation of energy homeostasis, nutrient metabolism (protein, carbohydrates, and lipid), intestinal health, and immunity in humans and animals [15,18]. In our preliminary experiments, the addition of 2.5 mg/mL isoleucine to drinking water improved the morphology and structure of rat intestinal mucosa [35]. In this study, we found that the addition of 1% Ile to the diet also significantly increased the duodenal villus height of weaned piglets and significantly elevated the jejunal villus height reduced by RV infection. The overall effect of adding 1% Ile was better than that of 0.5% Ile. The results reveal that the addition of Ile to the diet promoted intestinal villi growth and alleviated the negative effects of rotaviruses on the morphology and structure of the intestinal tract of weaned piglets to a certain extent.

In terms of small intestinal immunity, the intestinal mucosal barrier is the first line of defense against endogenous and exogenous stimuli and pathogenic bacteria. The mucus of the intestinal mucosal barrier is mainly formed and secreted by goblet cells [36]. There is increasing evidence that certain enteropathogenic viruses, such as porcine epidemic diarrhea virus [37], murine astrovirus [38], and human enterovirus 71 [39], can inhibit the formation of goblet cells and affect their function. For example, porcine delta coronavirus infection severely damages the intestinal mucosal barrier by activating the Notch signaling pathway and inhibiting goblet cell differentiation and mucus secretion [40]. Goblet cells and mucins play an important role in actively defending against rotavirus infection. The number of goblet cells was significantly reduced in the duodenum and jejunum of rotavirus-infected mice [41]. In addition, porcine epidemic diarrhea virus (PEDV) infection reduced the number of goblet cells in the small intestine of piglets and resulted in a small number of epithelial goblet cells being virus-positive, suggesting that PEDV infection possibly leads to an impaired mucus layer and increased susceptibility to secondary enteric bacterial infection [42]. In this study, we found that RV infection significantly reduced the number of goblet cells throughout the small intestinal segments of weaned piglets. Notably, Ile restored the goblet cell counts reduced by RV infection. These findings suggest that RV infection inhibited the mucus production by goblet cells in the small intestine, while Ile alleviated this negative effect.

Tight junctions consist of multiple transmembrane proteins that connect epithelial cells. Intestinal tight junction proteins such as Claudin and Occludin play important roles in the formation and function of the intestinal mechanical barrier [43,44]. In the intestine, the transmembrane mucin Muc1 is an important member of the physical barrier, influencing epithelial cell morphology and receptor function [45]. In this study, 1% Ile increased the expression of Claudin 1 in the duodenum and jejunum, Occludin in the jejunum, and Muc 1 in the duodenum and jejunum of RV-infected piglets. This may be due to the fact that Ile enhances the immunity of weaned piglets, inhibiting the invasion of RV into small intestine epithelial cells and alleviating the damage to the tight junction mechanical barrier caused by RV infection, thus ensuring normal barrier function.

Enzyme activity in the digestive tract is an important factor influencing intestinal health and digestibility [46]. Different breeds of pigs exhibit significant differences in growth performance and small intestinal development, mainly due to differences in digestive enzymes and nutrient transporter proteins [47]. Various digestive enzymes are synthesized in the intestines and secreted into the intestinal lumen primarily by the pancreas, with their levels regulated by nutrient intake. A previous study mentioned that the duodenal instillation of Ile increases pancreatic secretion, especially α-amylase [48]. In addition, the activities of lipase and sucrase are key markers for evaluating intestinal epithelial development or maturation [49,50]. In this study, we found that isoleucine significantly increased the activity of amylase and sucrase in the ileum of the small intestines of the RV-uninfected weaned piglets. However, RV infection resulted in varying reductions in lipase, amylase, and sucrase activity. Furthermore, the 1% Ile-supplemented group with RV infection exhibited significantly increased duodenal lipase and ileal sucrase activity compared with the RV group. These results reveal that Ile has a modulating effect on small intestinal digestive enzymes and can attenuate the reduction in digestive enzyme activity caused by RV infection to a certain extent, although the specific mechanisms need further study.

Reactive oxygen species (ROS) play an important role in cellular signaling and homeostasis in vivo [51]. However, the excessive accumulation of ROS in organisms leads to the inactivation of proteins, enzymes, and phospholipids in cellular membranes and the activation of pro-inflammatory signaling pathways. The intestine is a primary target organ affected by ROS [52]. RV interactions with mitochondria lead to the increased production of ROS [27,53]. Thus, maintaining antioxidant capacity during RV infection is important for the inhibition of replication within viral vectors. In our previous study, we concluded that drinking water supplemented with an appropriate amount of Ile could improve serum antioxidant capacity in rats [54]. In this study, we found that Ile significantly increased the expression of T-AOC in the jejunum and the expression of GSH in the ileum of the uninfected group. Moreover, the addition of 1% Ile in the RV-infected group significantly increased the expression of SOD in the duodenum as well as the expression of GSH in the jejunum.

IFN-γ is a pro-inflammatory cytokine produced during intestinal inflammation and has an important role in the intestinal inflammatory response [55]. In rotavirus-infected piglets, the concentration of IFN-γ significantly increased compared with that in healthy piglets [29]. Additionally, the addition of L-Ile to the diet increased the levels of IFN-γ in the serum of RV-infected weaned piglets [56]. SIgA is a polymeric immunoglobulin composed of dimeric IgA, a J chain, and a secretory component, most of which are generated by the gut and exert effects in situ [31]. In this study, we found that RV infection significantly increased the IFN-γ expression in both the jejunum and ileum, confirming that viral infection activates the antiviral state of the weaned piglet. The addition of 1% Ile increased the expression of interferon in the duodenum of the RV-infected group, indicating that isoleucine enhances small intestinal mucosal immunity after RV infection by increasing duodenal IFN-γ expression. Furthermore, we also found that with the addition of 1% Ile, the expression of SIgA in the duodenum and jejunum was significantly higher in the infected group compared with the uninfected group. These results demonstrate that Ile could promote the secretion of pro-inflammatory factors and regulate the immune ability of the organism to a certain extent. However, the specific mechanisms by which Ile influences intestinal immunity in piglets require further investigation.

Defensins are antimicrobial peptides produced by leukocytes and epithelial cells, acting directly on virus particles and host cells while playing an important role as antiviral agents [57]. Our previous study showed that Ile treatment enhanced the expression of porcine β-defensin-1 (pBD-1), pBD-2, pBD-3, pBD-114, and pBD-129 in the jejunum and ileum [58]. In this study, we found that 1% Ile increased the expression of DEFβ1 in the ileum. RV infection decreased the expression of DEFβ1 in the jejunum and the expression of DEFβ2 in the duodenum. Furthermore, the addition of 1% Ile in the RV-infected group significantly increased the expression of DEFβ1 in the jejunum and increased the expression of DEFβ2 in the duodenum. These findings suggest that Ile may inhibit RV infection by increasing jejunal DEFβ1 expression and duodenal DEFβ2 expression.

## 5. Conclusions

In summary, the addition of 1% Ile to the diet promoted the healthy development of the intestinal mucosa. Following RV infection, Ile was able to restore the reduced villus height in the ileum and the number of goblet cells in both the duodenum and ileum to normal levels, while also improving intestinal epithelial tight junctions. Additionally, Ile increased the activity of lipase, amylase, sucrase, SOD, and GSH, as well as the expression of SIgA, DEFβ1, and DEFβ2 in various segments of the small intestine. Therefore, the inclusion of Ile in the diet can alleviate the effects of RV on intestinal morphology and barrier function, as well as the physiological functions of weaned piglets. Moreover, Ile has the ability to enhance the antioxidant capacity and immune function of weaned piglets to a certain extent.

## Figures and Tables

**Figure 1 animals-14-03146-f001:**
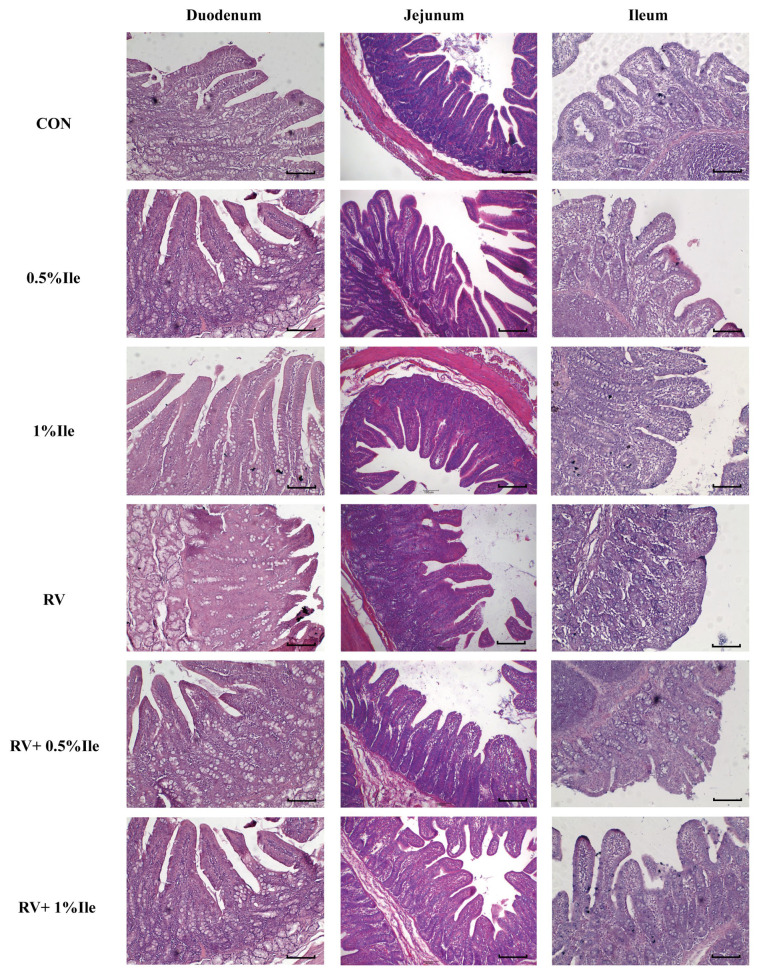
Effects of Ile supplementation on small intestinal morphology in RV-infected piglets. Representative histologic HE-stained micrographs. Scale bars are 200 μm.

**Figure 2 animals-14-03146-f002:**
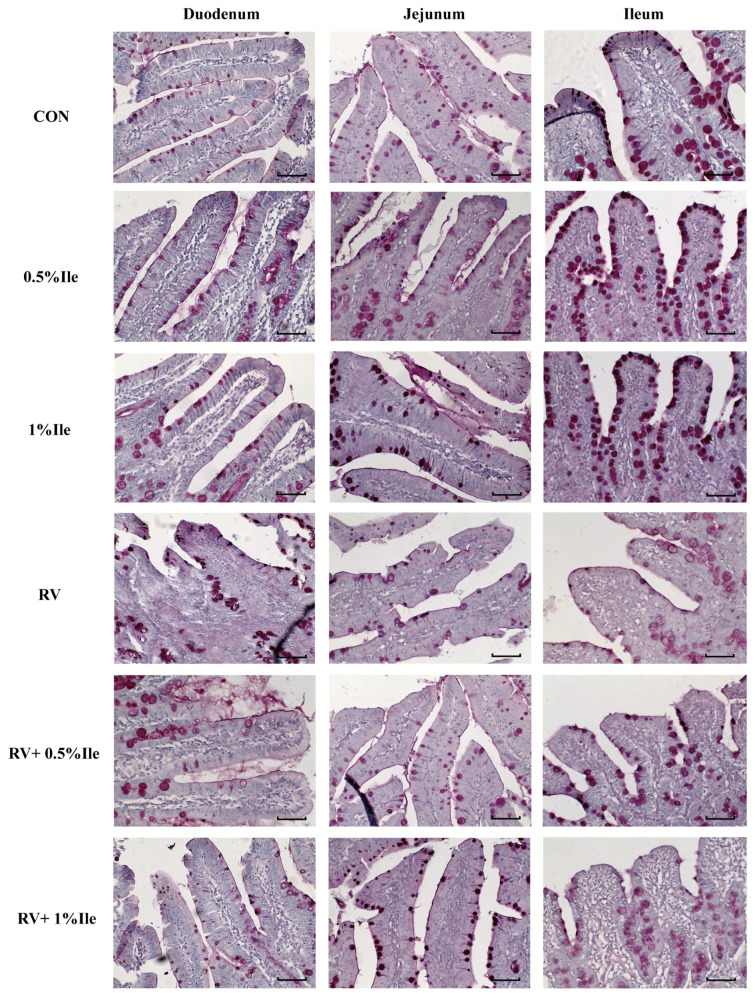
Effects of Ile supplementation on mucosal barrier function of the small intestine in RV-infected piglets. Representative histologic PAS-stained micrographs. Scale bars are 200 μm.

**Figure 3 animals-14-03146-f003:**
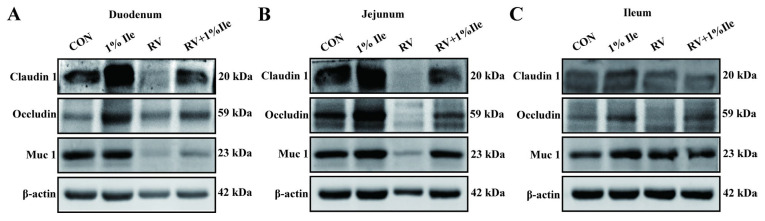
Effects of Ile supplementation on the mechanical barrier of the small intestinal mucosa in RV-infected weaned piglets. The expressions of Claudin 1, Occludin, and Muc 1 were measured in the duodenum (**A**), jejunum (**B**), and ileum (**C**) using Western blot analysis.

**Table 1 animals-14-03146-t001:** The animal experimental design.

	RV (−)			RV (5 mL)		
**Ile supplementation (%)**	0	0.5	1	0	0.5	1
**Groups**	CON	0.5% Ile	1% Ile	RV	RV + 0.5% Ile	RV + 1% Ile

**Table 2 animals-14-03146-t002:** Basal ration composition and nutrient levels.

Items	Groups		
	CON/ RV	0.5% Ile/ RV + 0.5% Ile	1% Ile/ RV + 0.5% Ile
**Ingredients (%)**			
Corn	61.27	61.11	60.95
Peeled soybean meal, 49% CP	6	6	6
Soybean protein concentrate, 65%	10	10	10
Fish meal, 64.5% CP	5	5	5
Deproteinized whey powder, 5.5% CP	5	5	5
Soybean oil	3	3	3
Corn starch	3.3	3.3	3.3
Glucose	2	2	2
Limestone	0.7	0.7	0.7
CaHPO_4_	1	1	1
NaCl	0.3	0.3	0.3
Chloride choline	0.15	0.15	0.15
Vitamin premix ^1^	0.03	0.03	0.03
Mineral premix ^2^	0.3	0.3	0.3
L-lysine·HCL	0.43	0.43	0.43
L-threonine	0.08	0.08	0.08
L-tryptophan	0.4	0.4	0.4
DL-methionine	0.06	0.06	0.06
L-valine	0.3	0.3	0.3
L-isoleucine	0	0.5	1
L-alanine	0.68	0.34	0
Total	100	100	100
**Calculated nutrient levels (%)**			
Crude protein	19.79	19.79	19.79
Calcium	0.8	0.8	0.8
Total phosphorus	0.65	0.65	0.65
Available phosphorus	0.48	0.48	0.48
Na	0.21	0.21	0.21
Cl	0.33	0.33	0.33
Lysine	1.35	1.35	1.35
Threonine	0.79	0.79	0.79
Tryptophan	0.22	0.22	0.22
Methionine	0.39	0.39	0.39
Valine	0.86	0.86	0.86
Isoleucine	0.72	1.22	1.72
Leucine	1.56	1.56	1.56
Leucine/isoleucine	217	128	91
**Metabolizable energy (MJ/kg)**	14.05	14.05	14.05

^1^ The vitamin premix provides the following per kg of diet: VA 9000 IU, VD3 3000 IU, VE 20.0 IU, VK3 3.0 mg, VB1 1.5 mg, VB2 4.0 mg, VB6 3.0 mg, VB12 0.2 mg, niacin 30.0 mg, pantothenic 15.0 mg, folic acid 0.75 mg, and biotin 0.1 mg. ^2^ The mineral premix provides the following per kg of diet: Fe 100 mg, Cu 6 mg, Zn 100 mg, Mn 4 mg, I 0.14 mg, and Se 0.3 mg.

**Table 3 animals-14-03146-t003:** Antibodies used in this study.

Antibodies	Supplier	Catalog No.
β-Actin Monoclonal antibody	Proteintech, Wuhan, China	66009-1-Ig
Claudin1 Polyclonal antibody		13050-1-AP
Occludin Polyclonal antibody		27260-1-AP
MUC-1 Rabbit pAb	Abclonal, Wuhan, China	A0333
HRP-conjugated Goat anti-Rabbit IgG (H + L)		AS014
HRP-conjugated Goat anti-Mouse IgG (H + L)		AS003

**Table 4 animals-14-03146-t004:** Kits for antioxidant and digestive enzyme activity assays.

Items	Supplier	Catalog No.	Function
Total antioxidant capacity (T-AOC) assay kit (ABTS method)	Nanjing Jiancheng Bioengineer Institute, Nanjing, China	A015-2-1	Measures total antioxidant capacity
Superoxide dismutase (SOD) assay kit (WST-1 method)		A001-3-1	Measures the activity of superoxide dismutase
Glutathione peroxidase (GSH-PX) assay kit (colorimetric method)		A015-2-1	Measures the activity of glutathione peroxidase
Lipase (LPS) assay kit		A054-2-1	Measures the activity of lipase
α-Amylase (AMS) assay kit		C016-1-1	Measures the activity of amylase
Sucrase assay kit		A082-2-1	Measures the activity of sucrase

**Table 5 animals-14-03146-t005:** Kits for cytokine expression assays.

Items	Supplier	Catalog No.	Function
Porcine IFN-γ ELISA kit	Dakome, Beijing, China	DKM2552A	Measures the expression of interferon gamma
Porcine SIgA ELISA kit		DKM9851A	Measures the expression of secretory immunoglobulin A
Porcine DEFβ1 ELISA kit		DKM0031PA	Measures the expression of defensin beta 1
Porcine DEFβ2 ELISA kit		DKM0035PA	Measures the expression of defensin beta 2

**Table 6 animals-14-03146-t006:** Effects of Ile supplementation on small intestinal morphology in RV-infected piglets (n = 8).

Items	RV−			RV+			*p*-Value		
	CON	0.5% Ile	1% Ile	CON	0.5% Ile	1% Ile	Ile	RV	Ile × RV
**Duodenum**									
VH, μm	363.08 ± 30.61 ^bc^	424.15 ± 58.60 ^ab^	465.15 ± 54.35 ^a^	313.46 ± 28.87 ^c^	367.85 ± 21.24 ^bc^	392.71 ± 38.65 ^abc^	<0.01	<0.01	0.81
CD, μm	268.16 ± 55.60	272.68 ± 42.56	250.91 ± 44.67	305.45 ± 31.8	274.62 ± 9.48	299.67 ± 22.47	0.74	0.07	0.49
VH-to-CD ratio	1.40 ± 0.27 ^b^	1.56 ± 0.14 ^ab^	1.87 ± 0.14 ^a^	1.03 ± 0.10 ^c^	1.34 ± 0.10 ^bc^	1.32 ± 0.23 ^bc^	<0.01	<0.01	0.21
**Jejunum**									
VH, μm	318.91 ± 67.11 ^ab^	375.96 ± 48.74 ^a^	320.94 ± 62.54 ^ab^	239.90 ± 36.69 ^b^	237.02 ± 29.47 ^b^	308.89 ± 53.03 ^ab^	0.23	<0.01	0.02
CD, μm	219.65 ± 51.15 ^c^	235.69 ± 35.93 ^c^	251.49 ± 38.37 ^bc^	316.02 ± 66.53 ^ab^	290.41 ± 43.67 ^abc^	345.50 ± 26.61 ^a^	0.14	<0.01	0.46
VH-to-CD ratio	1.46 ± 0.07 ^ab^	1.60 ± 0.08 ^a^	1.29 ± 0.25 ^b^	0.78 ± 0.14 ^c^	0.83 ± 0.14 ^c^	0.91 ± 0.21 ^c^	0.18	<0.01	0.02
**Ileum**									
VH, μm	330.68 ± 41.11 ^ab^	346.27 ± 38.43 ^a^	348.04 ± 54.77 ^a^	223.58 ± 30.72 ^c^	256.86 ± 62.47 ^bc^	311.49 ± 13.57 ^ab^	0.06	<0.01	0.21
CD, μm	224.92 ± 24.55	207.09 ± 12.53	211.05 ± 26.45	204.24 ± 45.95	223.58 ± 31.05	215.27 ± 14.97	0.98	>0.99	0.36
VH-to-CD ratio	1.47 ± 0.02 ^a^	1.67 ± 0.14 ^a^	1.64 ± 0.09 ^a^	1.12 ± 0.14 ^b^	1.14 ± 0.18 ^b^	1.45 ± 0.12 ^a^	<0.01	<0.01	0.02

RV−, rotavirus-uninfected; RV+, rotavirus-infected; CON, L-alanine-supplemented diet; Ile, L-isoleucine-supplemented diet. VH, villus height; CD, crypt depth. ^a,b,c^ In the same row, values with different letter superscripts mean a significant difference (*p* < 0.05).

**Table 7 animals-14-03146-t007:** Effects of Ile supplementation on goblet cell number of the small intestine in RV-infected piglets (n = 8).

Items	RV−			RV+			*p*-Value		
	CON	0.5% Ile	1% Ile	CON	0.5% Ile	1% Ile	Ile	RV	Ile × RV
Duodenum GCs number/villus	13.83 ± 2.79 ^a^	15.00 ± 1.50 ^a^	13.33 ± 1.75 ^a^	6.83 ± 1.94 ^b^	9.83 ± 1.47 ^b^	10.00 ± 1.41 ^b^	0.03	<0.01	0.06
Jejunum GCs number/villus	15.83 ± 3.37	17.40 ± 3.05	16.50 ± 3.94	13.00 ± 1.63	13 ± 2.37	13.83 ± 2.99	0.81	0.05	0.77
Ileum GCs number/villus	15.33 ± 0.82 ^a^	14.83 ± 2.14 ^a^	14.33 ± 3.14 ^a^	6.17 ± 2.32 ^c^	10.83 ± 1.17 ^b^	9.33 ± 1.63 ^bc^	0.06	<0.01	0.01

RV−, rotavirus-uninfected; RV+, rotavirus-infected; CON, L-alanine-supplemented diet; Ile, L-isoleucine-supplemented diet. GC, goblet cell. ^a,b,c^ In the same row, values with different letter superscripts mean a significant difference (*p* < 0.05).

**Table 8 animals-14-03146-t008:** Effects of Ile supplementation on the mechanical barrier of the small intestinal mucosa in RV-infected weaned piglets (n = 8).

Items	RV−		RV+		*p*-Value		
	CON	1% Ile	CON	1% Ile	Ile	RV	Ile × RV
**Duodenum**							
Claudin 1	1.00 ± 0.06 ^b^	1.37 ± 0.10 ^a^	0.73 ± 0.03 ^c^	1.13 ± 0.06 ^b^	<0.01	<0.01	0.78
Occludin	1.00 ± 0.25 ^b^	1.74 ± 0.10 ^a^	1.01 ± 0.18 ^b^	1.38 ± 0.20 ^ab^	<0.01	0.14	0.12
Muc 1	1.00 ± 0.05 ^a^	1.08 ± 0.04 ^a^	0.39 ± 0.02 ^c^	0.51 ± 0.03 ^b^	<0.01	<0.01	0.30
**Jejunum**							
Claudin 1	1.00 ± 0.08 ^a^	1.06 ± 0.03 ^a^	0.33 ± 0.06 ^c^	0.69 ± 0.03 ^b^	<0.01	<0.01	<0.01
Occludin	1.00 ± 0.04 ^a^	1.14 ± 0.10 ^a^	0.55 ± 0.14 ^b^	0.95 ± 0.08 ^a^	<0.01	<0.01	0.05
Muc 1	1 ± 0.01 ^a^	1.09 ± 0.05 ^a^	0.47 ± 0.09 ^b^	0.98 ± 0.10 ^a^	<0.01	<0.01	<0.01
**Ileum**							
Claudin 1	1.00 ± 0.05 ^b^	1.57 ± 0.16 ^a^	0.81 ± 0.05 ^b^	0.77 ± 0.03 ^b^	<0.01	<0.01	<0.01
Occludin	1.00 ± 0.03 ^b^	1.31 ± 0.09 ^a^	1.07 ± 0.05 ^b^	0.91 ± 0.10 ^b^	0.12	0.01	<0.01
Muc 1	1.00 ± 0.12 ^b^	1.42 ± 0.05 ^a^	1.08 ± 0.10 ^b^	1.08 ± 0.08 ^b^	<0.01	0.04	<0.01

RV−, rotavirus-uninfected; RV+, rotavirus-infected; CON, L-alanine-supplemented diet; Ile, L-isoleucine-supplemented diet. Data show the results of gray-scale value quantification, normalized by β-actin (relative). ^a,b,c^ In the same row, values with different letter superscripts mean a significant difference (*p* < 0.05).

**Table 9 animals-14-03146-t009:** Effects of Ile supplementation on digestive enzymes of the small intestinal mucosa in the RV-infected weaned piglets (n = 8).

Items	RV−		RV+		*p*-Value		
	CON	1% Ile	CON	1% Ile	Ile	RV	Ile × RV
**Duodenum**							
Lipase, U/g prot	20.45 ± 3.14 ^a^	20.70 ± 2.68 ^a^	14.69 ± 2.33 ^b^	19.63 ± 3.28 ^a^	0.04	0.01	0.06
Amylase, U/mg prot	0.10 ± 0.03	0.11 ± 0.03	0.08 ± 0.04	0.08 ± 0.03	0.71	0.08	0.71
Sucrase, U/mg prot	67.97 ± 17.58 ^ab^	78.80 ± 13.88 ^a^	32.00 ± 18.59 ^c^	49.27 ± 16.00 ^bc^	0.05	<0.01	0.64
**Jejunum**							
Lipase, U/g prot	19.58 ± 1.68 ^ab^	21.64 ± 2.03 ^a^	17.07 ± 2.14 ^bc^	16.10 ± 2.05 ^c^	0.51	<0.01	0.08
Amylase, U/mg prot	0.12 ± 0.02 ^a^	0.15 ± 0.02 ^a^	0.07 ± 0.02 ^b^	0.11 ± 0.04 ^ab^	<0.01	<0.01	0.65
Sucrase, U/mg prot	145.82 ± 20.92 ^a^	156.18 ± 28.54 ^a^	109.97 ± 19.69 ^b^	127.50 ± 37.65 ^ab^	0.23	0.01	0.75
**Ileum**							
Lipase, U/g prot	14.23 ± 2.78 ^ab^	16.61 ± 2.47 ^a^	11.34 ± 1.11 ^b^	14.49 ± 1.34 ^ab^	<0.01	0.01	0.65
Amylase, U/mg prot	0.10 ± 0.01 ^b^	0.14 ± 0.01 ^a^	0.07 ± 0.01 ^c^	0.10 ± 0.01 ^b^	<0.01	<0.01	0.24
Sucrase, U/mg prot	101.24 ± 22.50 ^b^	139.25 ± 26.20 ^a^	54.52 ± 13.29 ^c^	91.13 ± 17.06 ^b^	<0.01	<0.01	0.93

RV−, rotavirus-uninfected; RV+, rotavirus-infected; CON, L-alanine-supplemented diet; Ile, L-isoleucine-supplemented diet. ^a,b,c^ In the same row, values with different letter superscripts mean a significant difference (*p* < 0.05).

**Table 10 animals-14-03146-t010:** Effects of Ile supplementation on antioxidant expression of the small intestinal mucosa in RV-infected weaned piglets (n = 8).

Items	RV−		RV+		*p*-Value		
	CON	1% Ile	CON	1% Ile	Ile	RV	Ile × RV
**Duodenum**							
T-AOC, U/mg prot	0.87 ± 0.10	0.86 ± 0.11	0.85 ± 0.09	0.92 ± 0.06	0.36	0.54	0.23
SOD, U/mg prot	5.15 ± 1.04 ^b^	7.35 ± 1.59 ^a^	4.56 ± 0.49 ^b^	7.11 ± 1.29 ^a^	<0.01	0.33	0.68
GSH, U/mg prot	0.87 ± 0.54 ^ab^	1.24 ± 0.36 ^a^	0.72 ± 0.31 ^b^	1.08 ± 0.16 ^ab^	0.01	0.24	0.97
**Jejunum**							
T-AOC, U/mg prot	0.73 ± 0.12 ^b^	0.97 ± 0.10 ^a^	0.69 ± 0.13 ^b^	0.81 ± 0.16 ^ab^	0.01	0.28	0.83
SOD, U/mg prot	6.87 ± 1.31	6.41 ± 1.98	6.22 ± 2.44	8.00 ± 3.14	0.43	0.57	0.18
GSH, U/mg prot	1.92 ± 0.38 ^ab^	2.14 ± 0.74 ^ab^	1.37 ± 0.64 ^b^	2.2 ± 0.52 ^a^	0.03	0.31	0.21
**Ileum**							
T-AOC, U/mg prot	0.81 ± 0.10 ^b^	0.92 ± 0.04 ^a^	0.80 ± 0.05 ^b^	0.82 ± 0.08 ^b^	0.02	0.04	0.09
SOD, U/mg prot	3.19 ± 2.36 ^ab^	4.86 ± 1.61 ^a^	2.57 ± 0.47 ^b^	2.46 ± 1.21 ^b^	0.17	0.01	0.12
GSH, U/mg prot	1.46 ± 0.54 ^b^	3.07 ± 0.74 ^a^	0.46 ± 0.23 ^c^	0.95 ± 0.56 ^bc^	<0.01	<0.01	0.05

RV−, rotavirus-uninfected; RV+, rotavirus-infected; CON, L-alanine-supplemented diet; Ile, L-isoleucine-supplemented diet. T-AOC, total antioxidant capacity; GSH, glutathione peroxidase; SOD, superoxide dismutase. ^a,b,c^ In the same row, values with different letter superscripts mean a significant difference (*p* < 0.05).

**Table 11 animals-14-03146-t011:** Effects of Ile supplementation on cytokine expression of the small intestinal mucosa in RV-infected weaned piglets (n = 8).

Items	RV−		RV+		*p*-Value		
	CON	1% Ile	CON	1% Ile	Ile	RV	Ile × RV
**Duodenum**							
IFN- γ, pg/mg prot	45.64 ± 12.56 ^b^	46.72 ± 13.67 ^b^	48.47 ± 5.51 ^b^	69.94 ± 14.90 ^a^	0.04	0.02	0.07
SIgA, μg/mg prot	2.31 ± 0.37 ^b^	2.50 ± 0.58 ^b^	2.08 ± 0.24 ^b^	3.42 ± 0.73 ^a^	<0.01	0.12	0.01
**Jejunum**							
IFN- γ, pg/mg prot	40.50 ± 11.7 ^b^	48.60 ± 12.35 ^b^	86.95 ± 11.70 ^a^	82.00 ± 17.55 ^a^	0.78	<0.01	0.25
SIgA, μg/mg prot	2.23 ± 0.73 ^b^	2.36 ± 0.21 ^b^	2.64 ± 0.83 ^b^	3.70 ± 0.65 ^a^	0.08	<0.01	0.20
**Ileum**							
IFN- γ, pg/mg prot	58.50 ± 13.95 ^b^	59.85 ± 13 ^b^	83.80 ± 16.65 ^a^	86.35 ± 6.30 ^a^	0.72	<0.01	0.91
SIgA, μg/mg prot	2.04 ± 0.29	2.50 ± 0.54	2.22 ± 0.63	2.79 ± 0.48	0.06	0.36	0.83

RV−, rotavirus-uninfected; RV+, rotavirus-infected; CON, L-alanine-supplemented diet; Ile, L-isoleucine-supplemented diet. IFN-γ, interferon gamma; SIgA, secretory immunoglobulin A. ^a,b^ In the same row, values with different letter superscripts mean a significant difference (*p* < 0.05).

**Table 12 animals-14-03146-t012:** Effects of Ile supplementation on defensin expression of the small intestinal mucosa in RV-infected weaned piglets (n = 8).

Items	RV−		RV+		*p*-Value		
	CON	1% Ile	CON	1% Ile	Ile	RV	Ile × RV
**Duodenum**							
DEFβ1, pg/mg prot	59.36 ± 8.80 ^ab^	67.53 ± 8.72 ^a^	49.88 ± 6.10 ^b^	50.94 ± 7.07 ^b^	0.16	<0.01	0.28
DEFβ2, pg/mg prot	68.09 ± 6.65 ^b^	77.57 ± 6.43 ^a^	54.64 ± 8.09 ^c^	72.83 ± 7.31 ^ab^	<0.01	0.01	0.15
**Jejunum**							
DEFβ1, pg/mg prot	32.29 ± 5.33 ^a^	33.54 ± 4.19 ^a^	23.36 ± 4.69 ^b^	31.96 ± 4.84 ^a^	0.02	0.01	0.08
DEFβ2, pg/mg prot	68.06 ± 9.57	70.39 ± 7.94	61.16 ± 4.77	69.06 ± 8.50	0.13	0.22	0.40
**Ileum**							
DEFβ1, pg/mg prot	18.29 ± 5.06 ^b^	26.45 ± 6.01 ^a^	18.80 ± 3.58 ^ab^	20.01 ± 3.93 ^ab^	0.03	0.14	0.09
DEFβ2, pg/mg prot	55.69 ± 7.73	60.28 ± 5.99	57.75 ± 9.10	61.15 ± 4.18	0.18	0.61	0.84

RV−, rotavirus-uninfected; RV+, rotavirus-infected; CON, L-alanine-supplemented diet; Ile, L-isoleucine-supplemented diet. DEFβ1, defensin beta 1; DEFβ2, defensin beta 2. ^a,b,c^ In the same row, values with different letter superscripts mean a significant difference (*p* < 0.05).

## Data Availability

The original contributions presented in this study are included in this article/Appendix A; further inquiries can be directed to the corresponding authors.

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
