# Peer review of "Isoleucine Enhanced the Function of the Small Intestinal Mucosal Barrier in Weaned Piglets to Alleviate Rotavirus Infection"

_animals, 2024, doi:10.3390/ani14213146_

Round 1
Reviewer 1 Report
Comments and Suggestions for Authors
In the present study, the authors aim to preliminarily investigate the effects of Ile on the intestinal tissue structure, digestive capacity and immune function in RV infected weaned piglets. This work has certainly a great potential to be interesting for a general audience, after modifying some minor problems.
1. Only two gradients are set in the experiment. Is it representative enough? Can the experiment time of 5 days only represent short-term effects?
2. It will be more intuitive to use small letters to mark the significance of histogram in the article.
Comments on the Quality of English LanguageMinor editing of English language required.
Reviewer 2 Report
Comments and Suggestions for Authors
This is an interesting and well-presented study but lacks clear proof that isoleucine has a role in promoting resistance or recovery from enteric virus infection. There is no data presented of the impact of Ile on symptoms of the infected piglets as at 5 days post infection (dpi), all piglets were euthanized and tissue samples were collected. Why after only 5 days needs to be explained, as does if the rotavirus infected animals were symptomatic. I wonder about the significance of the Ile protection as the overall data suggests that the Ile protection is not that strong numerically.
What is the purpose of this study if isoleucine was not able to reduce the symptom severity of the rotavirus infected animals? Without this information the biological significance of the Ile supposed protection to the gut epithelia shown in Fig 1A is less impactful (and frankly all the tissue sections look similar in morphology).
Reviewer 3 Report
Comments and Suggestions for Authors
Interesting manuscript. The introduction is well written, gives enough background. In materials and methods animal housing information are missing. Results are not presented according to the statistical method described, therefore the comparisons are not really valid. This makes the suggested conclusions questionable. The improvement of the presentation of results is a must.
L75: Piglets were weaned at 21d? Gender distribution? Provide clear information. What was the average live weight of the piglets? Housing of piglets? Individually or in groups? How feed and water was provided? What was the feed intake? Was the basal experimental feed fed to the piglets before weaning? What was fed before weaning?
L81: How the RV was obtained, how was the solution prepared? Provide source information.
L82: certainly not saline was inoculated, but saline solution. What concentration of saline solution?
table 1: second hair line should continue
Ile (%): more precise: Ile supplementation (%) - because the diet itself contained some Ile.
Table 2: What is Peeled soybean merl? I did not found such a product. Provide manufacturer, city of headquarter and country.
Whey powder: is it Whey protein powder? It certainly has more than 3% of CP.
As L-Ala was exchanged to l-Ile on N basis, the levels needs to be presented for all experimental diets, not only for the basal.
As it is important, Leu/Ile ratio has to be provided for all diets.
Metabolic energy does not have the unit of %
L97 Provide exact details of procedure.
L98 How the same position was ensured?
L101: Was the lumen washed to remove chymus?
L104: Are these the same samples described in paragraph 2.3?
L135: Authors should test and report interaction effects as well.
L145: Software used, and method of measurement is not described in materials and methods.
Figure 1BCD: This type of significant difference coding reveals that the data were analyzed as one way anova (six treatments, not two times three), that is the reason why no interaction effect is reported. This makes the results much less interpretable. Correct it, in all relevant figures.
L165: Method of determination is not provided.
L16: ileum (no capital letter)
Figure 3, 4, 5, 6: why only two Ile treatment is analysed? This should be discussed in statistical analyses. This again not the result of a two way anova.
L210-212: This is already a discussion.
I do suggest to combine results with discussion.
Round 2
Reviewer 2 Report
Comments and Suggestions for Authors
The manuscript has been improved and the author's response to my queries over Ile improving symptoms of the infection have been addressed. Thank you.
Author Response
Thank you very much for your comments and professional advice. These opinions help to improve academic rigor of our article.
Reviewer 3 Report
Comments and Suggestions for Authors
The authors answered the concerns and made changes accordingly.
I have detected a few issues:
L92: Piglets had free access to feed and water. Watered sounds to me like watering the plants.
Table 2: Be precise: Deproteinized whey powder, Metabolizable energy
L116-117: How the sections border of small intestine was determined?
Figures: I am still not satisfied with the presentation (barcharts). Normally, when the interaction is not significant, we present the means of one treatment over the group means of the other treatment. Like presenting the RV effect by averaging the Ile treatments, and presenting the effect of Ile treatments as the average of RV groups at each level. When the interaction is significant, than it means that the response to Ile supplementation depends on RV status. Than we normally present the group means as a matrix (table), rows can represent RV status, columns the Ile supplementation, and we provide P value and RMSE for each row and column (one way anova). In that case significance of difference between means are marked by superscripts.
Figure 2: C: RV effect highly significant, but no group differences indicated.
Figure 2: D: Ile effect not significant, but group differences are marked. If the model is not significant, we do not mark significant differences even the range test shows it. Check every figure for such inconsistencies.
